# Population interest in and adequacy of the information on the safety of antineoplastic agents in the Spanish edition of Wikipedia

Seira Climent-Ballester[1], Pedro García-Salom[1], Javier Sanz-Valero [2]*

1 Pharmacy Service, Dr. Balmis General University Hospital, Alicante, Spain, 2 Carlos III Health Institute, National School of Occupational Medicine, Madrid, Spain

* fj.sanz@isciii.es

## Abstract

### Objective

To analyse the trend in use of the main antineoplastic agents (ANP) in Spain, to determine the association of this trend with the number of visits to the related pages in the Spanish edition of Wikipedia and to verify the existence of information aimed at reducing the associated risks of exposure to these drugs.

### Methods

This study had an ecological, descriptive cross-sectional design. The ANP for which more than 100,000 units were used per year in the Spanish Health System were included in the analyses. The trend in the use of these ANP and the number of visits to the pages for these ANP in Wikipedia were analysed using a regression model, and the correlation of these variables was evaluated. Fulfilment of the criteria related to medical-pharmaceutical information (MPI) and safety measure information (SMI) was determined.

### Results

An increasing trend in the use of ANP was observed for the 9 ANP included in this study: paclitaxel, fluorouracil, azacitidine, oxaliplatin, rituximab, carboplatin, doxorubicin, etoposide, cyclophosphamide, and fluorouracil, which were the most commonly used ANP in the study period. Visits to the Wikipedia pages for the 9 ANP showed a decreasing trend, with an inverse relationship between use and visits to the related Wikipedia pages. Regarding MPI criteria, only the indication/use was included in all ANP pages, and no more than 30% of the remaining criteria were met, with the exception of rituximab, for which 50% of the remaining criteria were met. The SMI criteria were not fulfilled by the ANP pages; effects on fertility were included in 2 (22%) ANP pages and effects on pregnancy were included in 4 (44%) ANP pages. A molecular identifier appeared in 7 of the ANP pages.

**Data availability statement:** All relevant data are within the manuscript.

**Funding:** The author(s) received no specific funding for this work.

**Competing interests:** The corresponding author, on behalf of herself and all her fellow authors, declares that there are no potential conflicts of interest related to this article.

## Conclusions

The consumption of ANP increased, whereas the population interest in visiting related Wikipedia pages decreased. Neither MFI nor SMI were readily available in the ANP articles (pages), including information on the risk of exposure to these dangerous drugs or how to reduce this risk.

## Introduction

Cancer continues to be one of the leading causes of mortality worldwide, with approximately 9.7 million deaths related to neoplasms in 2022 according to data provided by the International Agency for Research on Cancer (IARC). The IARC estimated that, in that same year, approximately 20 million new cases of neoplasms were diagnosed (excluding nonmelanoma skin tumours) and that this number would increase in 2025 to 35.0 million [1].

Chemotherapy is a therapeutic strategy used to treat cancer. Currently, there is a wide arsenal of drugs with antineoplastic activity, and although therapies directed at specific cancer cell targets are becoming increasingly numerous, conventional chemotherapy continues to be the predominant method of treating cancer. These latter drugs are more nonspecific and generate a greater number of associated side effects because they act not only on tumour cells but also on healthy cells [2].

The toxicity of different antineoplastic agents (ANP) has a negative influence on the quality of life of patients, as well as on their morbidity and mortality. Adverse reactions in organs and tissues with high and rapid cell multiplication, such as hair follicles, bone marrow, and organs of the digestive tract and reproductive system, are well known. However, it is the nausea and vomiting induced by these medications that are the side effects that most concern patients with neoplasms [3,4]. Poor control of nausea and vomiting can lead to deterioration of the physical condition of patients, thus affecting their quality of life [5,6] and potentially leading to cycle delays or dose reductions in planned cancer treatment, which could affect treatment efficacy.

From a different perspective, the exposure of nurses, pharmacists, doctors, laboratory technicians, drug transport personnel, patients and their caregivers to ANP is a potential health hazard. In addition, these compounds are potential environmental pollutants if they are disposed of improperly, which makes them a matter of immense concern in terms of medical safety and even the environmental safety of patients with cancer [7]. Until a few years ago, drug treatment for cancer was carried out in specialized units within hospital facilities. However, today, technological advances and improvements in quality of care have made it possible for most chemotherapies to be administered on an outpatient basis [8].

Therefore, given the sociocultural impact of cancer diagnosis [9], the complexity of oncological disease and its treatments, and the high fear of patients regarding the side effects and dangers derived from their exposure to ANP [2], it is not surprising that information is sought on these ANP, even many times on Web 2.0.

Web 2.0 resources have led to a substantive change in the communication of knowledge, favouring its dissemination by allowing the expansion and permeability of knowledge at a very low cost. In addition, Web 2.0 has integrated into the current information society, and far from diminishing the information available to society, Web 2.0 has an increasing number of initiatives that have enhanced this information. It is not unreasonable to classify the Web as the main means of obtaining health information [10].

Many people claim to look for health information on the Web, even before consulting with professionals, and, in relation to searches about medicines on the internet, most of them are

due to a specific cause, such as episodic drug treatments [11]. Similarly, a good number of searches by the general public are made on "sensitive" topics related to health, so knowing the number of visits by a population of interest can be useful for making important decisions regarding health (therapies, treatments, medications, etc.) [12].

Wikipedia is an example of a Web 2.0 resource; it is an encyclopaedia edited in collaboration by volunteers from all over the world, has been available on the internet since 2003, and is currently the seventh most visited site on the internet [13]. Likewise, the articles (pages) it contains usually appear first in the results of main search engines, making Wikipedia the most visible and most consulted Web 2.0 resource [14].

Currently, it is essential that patients, families and caregivers seeking information about chemotherapy have access to information not only on the efficacy of the treatment but also on the chemical risk of exposure to these dangerous drugs (medication-related problems (MRPs)) and how to reduce, as much as possible, this exposure [15]. Therefore, the general population must learn and practice exposure prevention to reduce their risk of exposure to the minimum level possible.

However, although health education is generally indicated as part of the rational use of drugs to ensure treatment quality and patient safety [16], there is little awareness of the need to reduce the risk of exposure to ANP in the general population. To our knowledge, there is no campaign to prevent the risk of exposure to ANP for the general population.

Therefore, it is essential, on the one hand, to provide health education on the risk of exposure to ANP and, on the other hand, to control the information available in this regard on Web 2.0, as it is the main source of consultation for the general population.

Consequently, the objective of this study was to analyse the trend in the usage of the main ANP in Spain, to determine the association of this trend with the number of visits to the pages on these ANP in the Spanish edition of Wikipedia and to verify the existence of information tending to decrease the associated risks of ANP exposure.

## Materials and Methods

### Design

This was an ecological, descriptive cross-sectional study.

### Ethical aspects

In accordance with Law 14/2007 on biomedical research [17], the approval of the Ethics and Research Committee was not necessary, as secondary data were used.

### Source of data collection

The search terms under study were obtained from the list of active terms of Anatomical, Therapeutic, Chemical classification (ATC) Group L (antineoplastic and immunomodulating agents) and Subgroup L01 (Antineoplastic agents) [18].

The data on usage were provided by the General Subdirectorate of Pharmacy of the General Directorate of the Common Portfolio of Services of the National Health System (Spanish National System (SNS)) and Pharmacy of the Ministry of Health and corresponded to the usage in hospitals of the SNS public network, provided by the health services of autonomous communities from 2016 to 2022, as of March 26, 2024.

To determine the existence of these pages in Wikipedia and the number of visits to these pages, the Spanish edition of Wikipedia was accessed via the internet (https://es.wikipedia.org/). The final consultation date was May 24, 2024.

## ANP and Wikipedia pages studied

The ANP with the highest use were studied according to the data provided by the SNS; more than 100,000 units of the following ANP were used in the year 2022: paclitaxel, fluorouracil, azacitidine, oxaliplatin, rituximab, carboplatin, doxorubicin, etoposide and cyclophosphamide.

## Data collection and storage

The results obtained were downloaded in a standardized comma-separated values (CSV) format that allowed their subsequent storage in an Excel file. The quality control of this information was carried out by means of duplicate tables, with the correction of possible inconsistencies by consulting the original downloaded table.

## Comparison pattern

To determine the relevance of the information on the ANP in Wikipedia, the information contained in the online drug information centre of the Spanish Agency for Medicines and Health Products (Centro de Información Online De Medicamentos de la Agencia Española de Medicamentos y Productos Sanitarios (AEMPS) (CIMA)) was used (https://cima.aemps.es/).

CIMA, as a database of medicines produced by the AEMPS (the reference entity in Spain belonging to the Ministry of Health), presents rigorous information and is considered highly reliable since it operates under strict national and European regulations, and its procedures are aligned with the standards of the European Medicines Agency (EMA).

Likewise, the safety data sheets were consulted, according to Regulation (EC) no. 1907/2006, of each of the ANP studied [19–27].

The information obtained or compared had to pertain to humans.

## List of criteria

a)  Use of ANP:

Number of containers used per year in the hospitals of the public network of the SNS (since May 2022, data on the hospital-level use of the Autonomous Community of Galicia has not been collected because of technical problems that prevent data reporting to the Ministry of Health).

b)  Searches for the terms in Wikipedia:

Number of times the page for the ANP has been visited (from January 2016 to December 2022).

c)  Criteria related to medical-pharmaceutical information (MPI) (dichotomous: no/yes):

– Risk indication: Text or pictogram that indicates the classification of the ANP as a dangerous drug.
– Prescription/dispensing message: Indication of the need for the drug to be prescribed by a doctor and dispensed by prescription.
– Indication/use: Existence, or not, of information regarding therapeutic indications or situations in which the use of the ANP is indicated.
– Interactions: Signalling of a possible reaction between two or more drugs or between a drug and a food, a drink, a plant or a supplement.
– Information on renal failure (RF): What to do in case of RF or what precautions to take to prevent RF was reported.

– Information on hepatic failure (HF): What to do in case of HF or what precautions to take to prevent HF was reported.
– Adverse/side effects: Indications of possible unwanted harmful effects resulting from ANP treatment.
– Contraindications: Information on situations where ANP administration should be avoided.
– Precautions in veterinary use: Indications related to the risks for people from use in animals.
– Driving and using machinery: Effects of the ANP that may make it unsafe for patients to drive or operate other machinery.

d) Criteria related to security measures (dichotomous: no/yes):

– Safe use (warnings and precautions): Recommendations for self-protection and for the protection of the subjects who live with the patient being treated with the ANP to reduce the risks of exposure to the drug:
  ◦ Eye contact

  ◦ Skin contact

  ◦ Ingestion

  ◦ Inhalation

– Spillage: Precautions to be taken in case of accidental spillage of the product.
– Handling: Instructions for the safe handling of the ANP.
– Storage: Instructions for the correct and safe storage of the ANP.
– Personal protective equipment (PPE): The need to use any equipment or clothing that protects against one or more risk factors that may threaten the safety or health of people.
– Toxicological information: Information on how to avoid harmful effects on living organisms and actions to take in case of emergency.
– Effects on fertility: Warnings on the effects on human reproductive ability.
– Effects on pregnancy: Warnings about possible risks related to ANP use during pregnancy.
– Effects on lactation: Indications on risks for the mother or child during the lactation period.
– Waste management: Processes conducive to reducing possible risks caused by the management of unused drugs.
  ◦ Personal

  ◦ Environmental

– Transportation: Special security and precautionary measures that are necessary for the shipment and receipt of the ANP.

e) Existence of a link to a database that contains MPI or safety measure information (SMI) (dichotomous: no/yes):

– PubChem: Database of molecules, operated and maintained by the National Center for Biotechnology Information (NCBI) of the U.S. National Institutes of Health.
– DrugBank: A Canadian database that provides data on drugs that is dependent on the University of Alberta.
– ChemSpider: Database of chemical products owned by the UK Royal Society of Chemistry.

## Data analysis

The qualitative data are presented as the frequency and percentage, the quantitative data are presented as the mean or median, as measures of central tendency, and the maximum and minimum values were calculated.

To assess the relationship between the consumption of antineoplastic agents and the volume of queries to the same terms in the Spanish edition of Wikipedia, over time, they have been adjusted using linear regression analysis models.

The relationships between quantitative data were determined using Pearson's correlation analysis.

For data storage and statistical analyses, Statistical Package for the Social Sciences (IBM-SPSS), version 29 for Windows was used. The level of significance used was $\alpha = 0.05$.

All data were collected by two of the researchers of this study (SCB and JSV), in case of discrepancies they were reviewed by agreement of the three researchers of the present study.

## Results

### ANP use

From the data provided by the hospitals of the public network of the SNS, it was possible to determine the use of ANP for the years 2016 to 2022. During that period, more than 100,000 units of the 9 ANP under study were used (see Table 1).

The ANP most used was fluorouracil, with a mean use of 332 280.57 ± 25798.39 units, a median use of 348 834 units, a maximum use of 409,649 units in 2016 and a minimum use of 209,809 units in 2022. The ANP with the lowest average use was cyclophosphamide, with a mean use of 113,953 ± 4,646.86 units, a median use of 118,871 units, a maximum use of 128468 units in 2021 and a minimum use of 96159 units in 2016 (see Table 1).

The temporal evolution of ANP use, as determined via a regression model, revealed a significant increasing trend, except for the use of oxaliplatin and etoposide, as shown in Table 1.

### Number of visits to ANP Wikipedia pages

Analysis of number of visits to ANP Wikipedia pages allowed us to determine the annual number of visits to these pages in the Spanish edition of Wikipedia (see Table 2).

The ANP page most visited was that for rituximab, with a mean of 60061.00 ± 9832.51 visits, a median of 55432 visits, a maximum of 95028 visits in 2016 and a minimum of 31780

**Table 1. Use of antineoplastic agents by hospitals in the public network of the Spanish National Health System.**

| Antineoplastic agent | 2016 | 2017 | 2018 | 2019 | 2020 | 2021 | 2022 | $R^2$ [#] |
|---|---|---|---|---|---|---|---|---|
| Paclitaxel | 212053 | 215599 | 216052 | 250362 | 235308 | 252093 | 246487 | 0.73 |
| Fluorouracil | 409649 | 276852 | 332363 | 367065 | 348834 | 381392 | 209809 | 0.18 |
| Azacitidine | 117217 | 128739 | 133696 | 152526 | 161370 | 181091 | 173248 | 0.94 |
| Oxaliplatin | 160722 | 146630 | 143600 | 164453 | 145680 | 149894 | 154906 | 0.01[*] |
| Rituximab | 117351 | 115638 | 122467 | 136785 | 137440 | 148223 | 142468 | 0.87 |
| Carboplatin | 92217 | 96915 | 112883 | 121639 | 128199 | 138562 | 136976 | 0.95 |
| Doxorubicin | 91407 | 92922 | 110825 | 121681 | 136968 | 143209 | 125799 | 0.77 |
| Etoposide | 121785 | 126051 | 125359 | 134526 | 131990 | 134959 | 122781 | 0.15[*] |
| Cyclophosphamide | 96159 | 98400 | 112978 | 122325 | 120471 | 128468 | 118871 | 0.73 |

[#]Coefficient of determination;

[*]Not significant

visits in 2022. The least visited ANP page was that for azacitidine, with a mean of 3038.57 ± 352.68 visits, a median of 2602 visits; a maximum of 4577 visits in 2018, and a minimum of 1966 visits in 2022 (see Table 2).

The temporal evolution ANP page visits, as determined via a regression model, showed a significant decreasing trend except for visits to the azacitidine page, which showed a very moderately decreasing trend that was not significant (see Table 2).

## Association between the use of ANP and the number of visits to ANP pages in the Spanish edition of Wikipedia

The analysis of the correlation between ANP use and the number of visits to each ANP page in Wikipedia showed a strong inverse relationship for paclitaxel (R = -0.87 p = 0.010), rituximab (R = -0.97 p < 0.001), carboplatin (R = -0.95 p = 0.001) and doxorubicin (R = -0.82 p = 0.026), this indicates that as searches on Wikipedia increase, the consumption of these drugs tends to decrease, as shown in Fig 1. For the remaining ANP, no significant correlation was found between their use and the number of visits to their pages: fluorouracil (R = 0.38 p = 0.405), azacitidine (R = -0.39 p = 0.392), oxaliplatin (R = 0.26 p 0.569), etoposide (R = -0.34 p = 0.456) and cyclophosphamide (R = -0.70 p = 0.079).

### Criteria related to MPI

Analysis of the fulfilment of criteria related to the reduction in ANP-related risk showed that only one of these variables, indication/use, was included on the pages of the 9 (100%) ANP studied. Another adverse/secondary effect was included on the pages for 8 (88.88%) of the ANP (only the page for fluorouracil did not mention another adverse/secondary effect). The remaining variables related to MPI were not mentioned on even 50% of the ANP pages. On the other hand, the ANP page that fulfilled the most criteria was that for rituximab, fulfilling 5 (50%) of the 10 criteria. The remaining ANP pages fulfilled only 3 (30%) or 2 (20%) criteria (see Table 3).

### Criteria related to SMI

Analysis of the fulfilment of criteria related to risk/safety information showed that the number of ANP pages for which these criteria were met was notably low. Only partial fulfilment of two criteria was verified: that of effects on pregnancy, information for which was included on the pages for 4 (44.44%) ANP (fluorouracil, oxaliplatin, carboplatin and doxorubicin), and that

**Table 2. Annual number of visits on pages related to antineoplastic agents in the Spanish edition of Wikipedia.**

| Antineoplastic agent | 2016 | 2017 | 2018 | 2019 | 2020 | 2021 | 2022 | $R^2$ # |
|---|---|---|---|---|---|---|---|---|
| Paclitaxel | 37141 | 36993 | 35159 | 27879 | 23096 | 20002 | 16456 | 0.96 |
| Fluorouracil | 33512 | 31952 | 35884 | 29797 | 25407 | 19146 | 17254 | 0.83 |
| Azacitidine | 2508 | 3070 | 4577 | 4023 | 2602 | 2524 | 1966 | 0.15* |
| Oxaliplatin | 11623 | 11317 | 10332 | 11372 | 8550 | 7073 | 6452 | 0.85 |
| Rituximab | 95028 | 87005 | 76000 | 55432 | 41246 | 33936 | 31780 | 0.96 |
| Carboplatin | 30060 | 37759 | 22807 | 21476 | 16412 | 12252 | 11413 | 0.83 |
| Doxorubicin | 41683 | 42390 | 40816 | 36434 | 28528 | 21759 | 17954 | 0.91 |
| Etoposide | 12818 | 13328 | 10755 | 9626 | 7997 | 6153 | 4959 | 0.97 |
| Cyclophosphamide | 46953 | 48169 | 46381 | 43511 | 27855 | 21773 | 17559 | 0.87 |

# Coefficient of determination;

*Not significant

of effects on fertility, information for which was included on the pages for only 2 (22.22%)

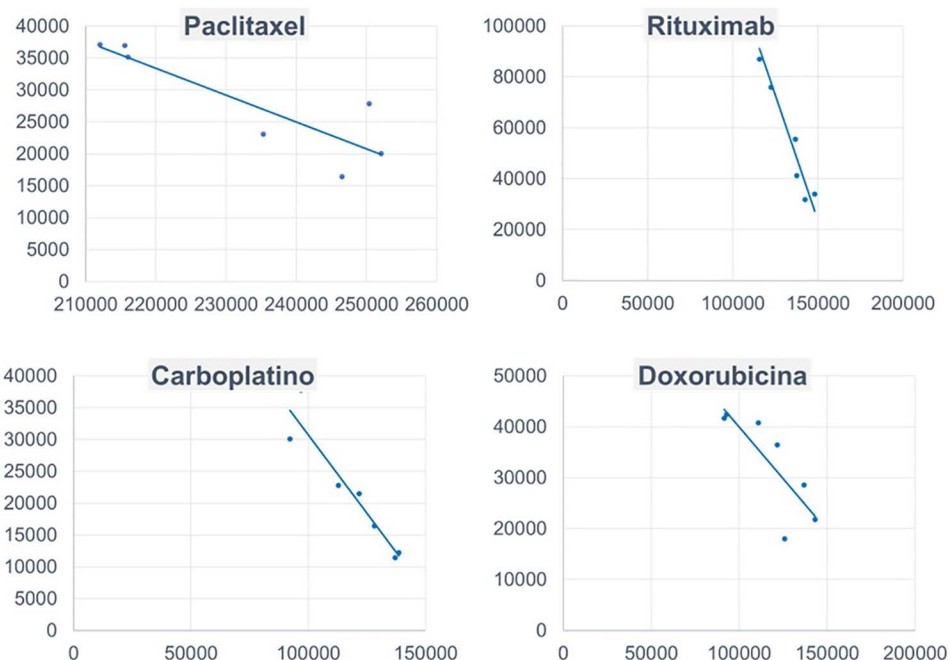

**Fig 1. Relationship between the use of antineoplastic agents (plotted on the horizontal axis) and the number of visits to the pages for these agents in the Spanish edition of Wikipedia (plotted on the ordinate axis).**

**Table 3. Fulfilment of criteria on including links to databases containing medical-pharmaceutical information related to the reduction in the risk of using anti-neoplastic agents.**

| | Antineoplastic agents | | | | | | | | | Rate of fulfilment |
|---|---|---|---|---|---|---|---|---|---|---|
| | Paclitaxel | Fluoro-uracil | Azaciti-dine | Oxal-iplatin | Ritux-imab | Carbo-platin | Doxoru-bicin | Etoposide | Clophos-phamide | |
| Risk indication | No | No | No | No | No | No | No | No | No | 0 (0.00%) |
| Prescription/dispensing | No | Yes | Yes | No | Yes | No | No | No | Yes | 4 (44.44%) |
| Indication/use | Yes | Yes | Yes | Yes | Yes | Yes | Yes | Yes | Si | 9 (100.00%) |
| Interactions | No | No | No | No | No | No | No | No | No | 0 (0.00%) |
| Kidney failure | No | No | No | No | No | No | No | No | No | 0 (0.00%) |
| Liver failure | No | No | No | No | Yes | No | No | No | No | 1 (11.11%) |
| Adverse/side effects | Yes | No | Yes | Yes | Yes | Yes | Yes | Yes | Yes | 8 (88.88%) |
| Contraindications | No | No | No | No | Yes | No | No | No | No | 1 (11.11%) |
| Driving and using machinery | No | No | No | No | No | No | No | No | No | 0 (0.00%) |
| Veterinary use | No | No | No | No | No | No | No | No | No | 0 (0.00%) |
| Total | 2 (20.00%) | 2 (20.00%) | 3 (30.00%) | 2 (20.00%) | 5 (50.00%) | 2 (20.00%) | 2 (20.00%) | 2 (20.00%) | 3 (30.00%) | |

**Table 4. Fulfilment of criteria on including links to databases containing safety measure information related to reducing the risk of using the antineoplastic agents.**

| | Antineoplastic agents | | | | | | | | | Rate of fulfilment |
|---|---|---|---|---|---|---|---|---|---|---|
| | aclitaxel | Fluoro-uracil | Azacit-idine | Oxal-iplatin | ituximab | Carbo-platin | Doxo-rubicin | Etopo-side | Cyclophos-phamide | |
| Eye contact | No | No | No | No | No | No | No | No | No | 0 (0.00%) |
| Skin contact | No | No | No | No | No | No | No | No | No | 0 (0.00%) |
| Ingestion | No | No | No | No | No | No | No | No | No | 0 (0.00%) |
| Inhalation | No | No | No | No | No | No | No | No | No | 0 (0.00%) |
| Accidental spillage | No | No | No | No | No | No | No | No | No | 0 (0.00%) |
| Manipulation | No | No | No | No | No | No | No | No | No | 0 (0.00%) |
| Storage | No | No | No | No | No | No | No | No | No | 0 (0.00%) |
| Personal protective equipment | No | No | No | No | No | No | No | No | No | 0 (0.00%) |
| Toxicological information | No | No | No | No | No | No | No | No | No | 0 (0.00%) |
| Effects on fertility | No | No | No | Yes | No | No | No | No | Yes | 2 (22.22%) |
| Effects on pregnancy | No | Yes | No | Yes | No | Yes | Yes | No | No | 4 (44.44%) |
| Effects on lactation | No | No | No | No | No | No | No | No | No | 0 (0.00%) |
| Waste management (danger to people) | No | No | No | No | No | No | No | No | No | 0 (0.00%) |
| Waste management (danger to the environment) | No | No | No | No | No | No | No | No | No | 0 (0.00%) |
| Transportation | No | No | No | No | No | No | No | No | No | 0 (0.00%) |
| Total | 0 (0.00%) | 1 (6.66%) | 0 (0.00%) | 2 (13.33%) | 0 (0.00%) | 1 (6.66%) | 1 (6.66%) | 0 (0.00%) | 1 (6.66%) | |

ANP (oxaliplatin and cyclophosphamide). Of note, the page for oxaliplatin included only 2 (13.33%) of the 15 criteria studied (Table 4).

## Existence of a link to a database containing MFI or SMI

Of the 9 ANP studied, only the page for fluorouracil included external links to the 3 databases that comprehensively include both MFI and the SMI. On the other hand, the pages for two of the ANP studied, rituximab and etoposide, did not include links to any of these 3 databases (see Table 5).

**Table 5. Existence of a link to a database that contains medical-pharmaceutical information or safety measure information for antineoplastic agents in the Spanish edition of Wikipedia.**

| Antineoplastic agent | PubChem | DrugBank | ChemSpider |
|---|---|---|---|
| Paclitaxel | Yes | Yes | No |
| Fluorouracil | Yes | Yes | Yes |
| Azacitidine | Yes | Yes | No |
| Oxaliplatin | Yes | Yes | No |
| Rituximab | No | No | No |
| Carboplatin | Yes | Yes | No |
| Doxorubicin | Yes | Yes | No |
| Etoposide | No | No | No |
| Cyclophosphamide | Yes | Yes | No |

## Discussion

The veracity of information about cytostatics is vital to ensure effective and safe treatments, to empower patients and to comply with ethical and legal standards. The inaccuracy or falsity of this information can have severe consequences, affecting both the health of patients and their caregivers and the health professionals who care for them. Thus, this work aimed to determine the general population's interest in the information concerning the most consumed ANP in Spain and to determine whether the existing information in the Spanish edition of Wikipedia could be useful for reducing the associated risks.

A growing use of ANP was found to be related to the increase in cancer incidence both in Spain and worldwide. In the year 2022, the use of ANP accounted for one-quarter of hospital pharmacy total expenditures [28].

Horn et al. highlighted that, despite the increasing incidence of cancer, oxaliplatin and etoposide use did not increase. Oxaliplatin, a first-line ANP for gastrointestinal tumours, is used in both neoadjuvant and adjuvant settings. However, the lack of an increase in oxaliplatin use cannot be explained by a decrease in the incidence of gastrointestinal cancers, because colorectal cancer is among the top three cancers in incidence and prevalence [29]. In contrast, the lack of an increase in etoposide use could be explained by a decrease in the incidence of one of its main indications, small cell lung cancer, which accounts for 15–20% of lung cancers, with a median survival of 10 months [30,31]. Although immunotherapy has been incorporated as a therapeutic option in patients with these cancers, it is done so as a combined treatment; that is, the standard chemotherapy is administered along with a monoclonal antibody [32]. The decrease in etoposide use may also be due to the evidence for long-lasting clinical responses when this ANP is used at low and stable doses [33].

In contrast to the increase in ANP use, the population of interest showed a decreasing trend in searches on these ANP. In line with these data, the study by Eheman et al. [34] revealed that only 1 in 5 cancer patients reported actively searching for information at the time of treatment, and Shea-Budgell et al. [35] concluded that although the internet was a source of information used by these patients, they preferred communication with their doctor or health professional because they considered them to be a more reliable source. In another study, cancer patients frequently searched the Web for health information after their doctor's appointment, perhaps because older people prefer direct consultation rather than using the internet to obtain health information [36]. In the study by Costas-Muniz et al. [37], only 2 out of 10 patients wanted additional information about their diagnosis or treatment.

One possible explanation is that although these drugs are more commonly used, they are often used in combination with other therapies (surgery, radiotherapy and chemotherapy), which makes it difficult for patients to fully understand the treatment they receive [38]. Epstein et al. [39] found that only 1 in 20 cancer patients who had survived for less than six months had an accurate understanding of their disease. In addition, chemotherapy causes important side effects (fatigue, nausea, vomiting, temporary hair loss, anaemia, infections, fertility problems, etc.), and these symptoms concern the patient and his or her environment and are the catalysts for most information searches [40,41].

The relationship between ANP use and searches for information on these ANP in the population interest showed in an inverse association. This trend has already been observed in other areas of study, especially when the symptoms or possible adverse effects are better known than the disease or the treatment [42]. This search for information is marked by the priority of responding to a need rather than informational interest. Law et al. [43] demonstrated this circumstance and concluded that the most frequent visits to medication pages were due to a specific cause, such as the possible consequences and MRPs of episodic pharmacological treatments. Thus, the work of Martínez-Aguilar et al. [44] showed that the

population's interest in MRPs was more inclined towards issues better known by the general population, such as overdose or contraindications.

Information on drugs is one of the most important aspects related to their use since having adequate information allows patients to obtain the best pharmacological treatment possible [45]. The misuse of medications by the patient is often due to their ignorance, which is generally due to a lack of information or understanding, misinterpretation or forgetfulness of the instructions that they received. Therefore, when a patient finds themselves alone and does not know how to use a medication on their own, on many occasions, they often end up consulting Wikipedia.

In this sense, the MFI included in the Spanish Wikipedia pages of the ANP studied was poor. There was an absence of information of real importance for those who consulted these pages. For example, no page presented information on what to do in case of RF, or no information was given on possible drug interactions [46].

Previous research on active medicinal ingredients, carried out in English, confirmed that important data, such as dosage, adverse effects and contraindications, are often lacking, resulting in serious damage to the health of patients [11]. Another study of 20 high-prescription drugs reported that the information on dosage and side effects was incomplete and inaccurate [47].

This situation is concerning given that U.S. doctors claim to use Wikipedia as one of their sources of information, and pharmacists themselves admit using Wikipedia in professional practice, recognizing that they are using Wikipedia increasingly more, even for educational purposes. For all of these health professionals, Wikipedia is the most commonly used Web resource, with the exception of Google [11].

There was no relationship of information presented on the Wikipedia page and the information in the Technical Data Sheets of the ANP studied. Technical Data Sheets include information on the need for dose adjustments in RF and HF, interactions, contraindications, effects on the ability to drive and use machines or waste management. This information is essential for the safe use of these ANP. The only information that all the pages included was the drug indication and its adverse effects.

Similarly, scarce data were provided in relation to SMI. This situation is very concerning given that therapeutic doses of ANP can clearly lead to harmful effects on patient health. Additionally, the possible adverse effects caused by chronic occupational exposure to low concentrations of these compounds are known, taking into account that these effects may be subclinical and may not be evident for years (or generations) after exposure. However, even in the absence of epidemiological data, the toxicity of ANP necessitates precautions to systematically minimize exposure [48]. For this reason, the lack of information on the risks of ANP use/manipulation, especially the lack of information on minimizing these risks, is remarkable.

In this study, no information on the risk of exposure to these ANP during breastfeeding was available; only 2 pages included information on the effects of the ANP on fertility, and 4 pages included information on the effects of the ANP on pregnancy.

Although it is known that there is no information on exposure to ANP in these populations because their use is contraindicated, there is also a lack of knowledge/misinformation on the potential risks of exposure in both healthcare professionals/staff and in caregivers and pregnant or breastfeeding family members. To date, these potential risks of exposure have not been studied, so a conservative attitude should be adopted, educating and warning of the dangers of exposure to these substances.

Furthermore, there was no information on safe use or occupational health, such as first aid, measures in case of accidental spillage, precautions in handling and storage, or exposure controls and individual protection. Moreover, no information on waste management or

environmental pollution was provided. Additionally, there was a lack of information on the safe use of ANP at home, where measures should be taken in cleaning, laundry and excreta handling. Even less information was available on precautions in the use of ANP in veterinary medicine.

The National Institute for Occupational Safety and Health (NIOSH), in collaboration with several university programs, carried out an intervention to improve the content of occupational safety and health in Wikipedia [49]. This collaboration led the European Agency for Safety and Health at Work (EU-OSHA) to develop a "wiki" tool, OSHwiki [50], which would allow the sharing of knowledge about safety and health at work. However, a query did not return information on ANP in general or on those studied in particular, although it did return an entry on "dangerous substances".

Along with concerns about the lack of both MFI and SMI, concern about the reliability of Wikipedia as a source of information in general is "advised", although not very prominently, by the encyclopaedia itself, since in its entry "Wikipedia: Wikipedia is not a reliable source", it is stated verbatim "Wikipedia is not a reliable source. Wikipedia can be edited by anyone at any time. This means that any information contained at any given time could be vandalism, a work in progress, or simply wrong" [51]. Furthermore, in another extensive entry on "Wikipedia reliability" there is a specific section on "Science and Medicine" where it is stated "Science and medicine are areas in which precision is of great importance and peer review is the rule. Although some Wikipedia content has undergone a form of peer review, most has not" [52]. The following page refers to a good number of academic works for and against the reliability of Wikipedia.

However, many people are unaware of these notices and may think that they receive adequate information. However, the spread of misleading MFI can cause significant harm to society and individuals, and, according to Hill et al. [53], it should be the responsibility of the content providers to solve this problem. Although Wikipedia is a free and collaboratively edited encyclopaedia, they must be responsible for the content that they disseminate, and it is no longer acceptable to hide behind the mantle of the "wiki".

The criterion of a link to specialized databases that would provide detailed MFI and SMI can also be considered unfulfilled, especially in for the two ANP pages in which there was no link. A link to PubChem alone could provide sufficient information, even with its pictogram portrayal of chemical safety. However, it is important to note that PubChem information is presented in English (the study was conducted in the Spanish edition of the Wikipedia). Therefore, many people who consult the Spanish edition of Wikipedia may not understand English or have enough training to consult these databases properly.

Studies such as those of Leithner et al. [54] and Kupferberg et al. [55], recommended that commonly used, non-peer-reviewed websites that provide health information, such as Wikipedia, should include links to reliable, high-quality, and well-maintained sources.

To achieve an adequate understanding of risk in the population, the development of timely, efficient and effective communication strategies, led by health personnel with sufficient background, that allows individuals to adequately use the means of communication at their disposal or integrate this information appropriately in teams made up of professionals from different specialties, is needed, particularly for those designated to take on these tasks [55].

Nevertheless, links to external web pages are critical to increasing the information provided to users with access to a variety of additional perspectives, data, and resources. These links help support and complement the information presented, increasing the credibility and authority of the content. Additionally, by offering users the opportunity to further explore a specific topic through external links, deeper learning and a fuller understanding are encouraged.

In summary, having verified the deficiencies in the existing information in the Spanish edition of Wikipedia, it is necessary to be very cautious when consulting the information on ANP in Wikipedia. This does not mean that users will not find accurate and valuable information on Wikipedia; the information will be most of the time. However, Wikipedia cannot guarantee the validity of the information found on their site [56].

## Possible limitations of the present study

The present work was limited to the study of the Spanish edition of Wikipedia, so it is possible that other editions will have different information from that presented here.

A limited number of ANP, the most used in Spain, were studied, and studying a larger number of ANP could provide more information.

The references included in the Wikipedia entries were not taken into account; it is possible that studying the references could provide more information on the current validity of the sources that support the contents, as has already been done in other areas of the health sciences [12].

It is possible that, as the years go by, Wikipedia will cease to be the reference Web tool, so this study would have to be updated in relation to the population's interest when it comes to carrying out consultations on the Internet.

On the other hand, the results offered by Google Trends could have been studied, but these are offered in percentage of search (relative search volume) with respect to the maximum search volume at a given time and, consequently, the relative search volume will fluctuate depending on the period studied. Therefore, it was decided to use Wikipedia data since these are real data on the number of searches and do not vary.

## Based on these findings, the following conclusions can be drawn

The use of ANP increased while the population interest in visiting related Wikipedia pages decreased. Neither MFI nor SMI were readily available in the ANP articles (pages), including information on the risk of exposure to these dangerous drugs or how to reduce this risk.

## Author contributions

**Conceptualization:** Seira Climent-Ballester, Javier Sanz-Valero.

**Data curation:** Seira Climent-Ballester, Javier Sanz-Valero.

**Formal analysis:** Seira Climent-Ballester, Pedro Garcia-Salom, Javier Sanz-Valero.

**Methodology:** Pedro Garcia-Salom, Javier Sanz-Valero.

**Supervision:** Javier Sanz-Valero.

**Validation:** Pedro Garcia-Salom, Javier Sanz-Valero.

**Visualization:** Seira Climent-Ballester, Javier Sanz-Valero.

**Writing – original draft:** Seira Climent-Ballester, Pedro Garcia-Salom, Javier Sanz-Valero.

**Writing – review & editing:** Seira Climent-Ballester, Pedro Garcia-Salom.

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
