## [Decision Letter · Decision Letter 0]

23 Sep 2024

PONE-D-24-35201Population interest in and adequacy of the information on the safety of antineoplastic agents in the Spanish edition of WikipediaPLOS ONE

Dear Dr. Sanz-Valero,

Thank you for submitting your manuscript to PLOS ONE. After careful consideration, we feel that it has merit but does not fully meet PLOS ONE’s publication criteria as it currently stands. Therefore, we invite you to submit a revised version of the manuscript that addresses the points raised during the review process.

**ACADEMIC EDITOR: **

Dear Authors,

The manuscript needs major revisions.

Respond point by point to the requests of the reviewers.

Kind regards.

We look forward to receiving your revised manuscript.

Kind regards,

Omar Enzo Santangelo

Academic Editor

PLOS ONE

Journal Requirements:

2. In your Methods section, please include additional information about your dataset and ensure that you have included a statement specifying whether the collection and analysis method complied with the terms and conditions for the source of the data.

Reviewers' comments:

Reviewer's Responses to Questions

**Comments to the Author**

1. Is the manuscript technically sound, and do the data support the conclusions?

Reviewer #1: No

Reviewer #2: Yes

2. Has the statistical analysis been performed appropriately and rigorously? 

Reviewer #1: No

Reviewer #2: Yes

3. Have the authors made all data underlying the findings in their manuscript fully available?

Reviewer #1: No

Reviewer #2: Yes

4. Is the manuscript presented in an intelligible fashion and written in standard English?

Reviewer #1: Yes

Reviewer #2: Yes

5. Review Comments to the Author

Reviewer #1: The initiative of the study is interesting in terms of therapeutic education but methodologically the work can be improved.

The fact that the study measures a negative correlation between the use of anticancer drugs and visits to Wikipedia pages does not seem enlightening to me. It is enough that over the years the referencing of websites changes for other 'Vidal' type websites to appear first in the results of the 'google' type web search before Wikipedia. So naturally Wikipedia will be less consulted.

Then the application of regression models (the nature of which we do not know) and correlation on such a small number of observations does not seem appropriate to me.

Reviewer #2: The manuscript was writtien on a very important topic where Web 2.0 became the main source of information in general, and the medical topics in specific. However, the authors heve to check that mentioning of the compund term prior is abbreviation to avoid readers' misleading (example, "AA" in rhe abstract).

Introduction section: There must be indication for recent data from IARC, as a legitimate source of updated statistics, where the number of neoplasms-related deaths became 10 million in 2022 (Refer to: https://www.iarc.who.int/news-events/new-report-on-global-cancer-burden-in-2022-by-world-region-and-human-development-level/) and the authors must avoid the use of referral references.

Besides, several paragraphs at the introduction section have no referral to any citated references, and thus the quality of those discussed information is questionable (paragraphs 2,5 and 9)

Materials and Methods: the number of topics being collected among popular Wikipedia pages were missing. The ethical aspects must be declared at first prior going into further details of the study methodology. The data quality as well as the statistical analysis must be performed by at least two authros / researchers in a blinded data selection, and handling the statistical analysis va validated software (as SPSS, STATA, R studio, SAS) was not clearly discussed in depth.

However, the authors must provide further information regarding how CIMA information center can be used as legitimate source and what type of validity being applied in the center to ensure its data accuracy.

Discussion: According to FDA Pregnancy category and drugs -related risks information, generally chemotherapy and AA in specific are considered category X medication; where studies in animals or humans have demonstrated positive evidence of human fetal risk based on adverse reaction data from investigational or marketing experience, and the risks involved in use of the drug in pregnant women clearly outweigh potential benefits. Thus, no such information regarding the use of AA in pregnant and breast feeding, or its impact on infants exposure.

6. PLOS authors have the option to publish the peer review history of their article (what does this mean? ). If published, this will include your full peer review and any attached files.

**Do you want your identity to be public for this peer review?** For information about this choice, including consent withdrawal, please see our Privacy Policy .

Reviewer #1: No

Reviewer #2: No

---

## [Author Response · Author response to Decision Letter 1]

20 Jan 2025

Point-by-point response to the reviewer’s comments

The authors thank the corrections and comments made. Suggestions for improvement provided by reviewers and editorial, have seemed us to be very appropriate and useful. We are confident that they have contributed to improving our article.

All modifications or additions made are listed in this document and are brightly-colored in the text of the manuscript for your easy viewing.

Revisions carried out

Reviewer #1

The initiative of the study is interesting in terms of therapeutic education but methodologically the work can be improved.

The fact that the study measures a negative correlation between the use of anticancer drugs and visits to Wikipedia pages does not seem enlightening to me. It is enough that over the years the referencing of websites changes for other 'Vidal' type websites to appear first in the results of the 'google' type web search before Wikipedia. So naturally Wikipedia will be less consulted.

Indeed, the results may change over the years and, therefore, this study should be updated. However, when selecting the Web 2.0 reference, we took into account the current results, as indicated in references 11 and 12, where we can find Wikipedia in position 7 in Similarweb and in position 5 in Semrush.

On the other hand, we could have studied the results offered by Google Trends, but these are offered in percentage of search (relative search volume) with respect to the maximum search volume at a given time and, consequently, the relative search volume will fluctuate depending on the period studied. Consequently, it was decided to use the data from Wikipedia as this is real data on the number of searches and does not vary.

As we consider that this explanation may be interesting, in accordance with the reviewer's warning, we have included it in the limitations of the study.

Then the application of regression models (the nature of which we do not know) and correlation on such a small number of observations does not seem appropriate to me.

We have modified the data analysis section in the methodology, explaining better the use of linear regression models and we have added a sentence in the results explaining, also, the data of the association between the consumption of antineoplastic drugs and the queries made to their terms in the Spanish edition of the Wikipedia.

Reviewer #2

The manuscript was written on a very important topic where Web 2.0 became the main source of information in general, and the medical topics in specific. However, the authors have to check that mentioning of the compound term prior is abbreviation to avoid readers' misleading (example, "AA" in the abstract).

We had forgotten to define the abbreviation AA in the abstract. However, taking advantage of the reviewer's recommendation, it is true that the abbreviation AA can lead to confusion as it is the abbreviation generally used for acetylsalicylic acid. On the other hand, neither the FDA (Food and Drug Administration) nor the EMA (European Medicines Agency) use a universally standardised abbreviation for antineoplastic agents. However, a review of the scientific literature has shown that ANP is the most commonly used abbreviation. We have therefore changed the abbreviation AA to ANP.

Introduction section: There must be indication for recent data from IARC, as a legitimate source of updated statistics, where the number of neoplasms-related deaths became 10 million in 2022 (Refer to: https://www.iarc.who.int/news-events/new-report-on-global-cancer-burden-in-2022-by-world-region-and-human-development-level/) and the authors must avoid the use of referral references.

Besides, several paragraphs at the introduction section have no referral to any citated references, and thus the quality of those discussed information is questionable (paragraphs 2,5 and 9)

− The first paragraph of the introduction has been modified taking into account the bibliographical citation provided by the reviewer.

− References have been added to the paragraphs indicated by the reviewer.

Materials and Methods: the number of topics being collected among popular Wikipedia pages were missing.

In the section ‘ANP and Wikipedia pages studied’, the topics studied have been specified both in terms of consumption and in Wikipedia.

The ethical aspects must be declared at first prior going into further details of the study methodology.

− The section on ethical issues has been placed at the beginning of the methodology, after the study design, as requested by the reviewer.

The data quality as well as the statistical analysis must be performed by at least two authors / researchers in a blinded data selection, and handling the statistical analysis va validated software (as SPSS, STATA, R studio, SAS) was not clearly discussed in depth.

− How the data was collected and the correction of discrepancies has been included in the ‘Data analysis’ section.

However, the authors must provide further information regarding how CIMA information center can be used as legitimate source and what type of validity being applied in the center to ensure its data accuracy.

− CIMA is the database of medicines produced by the Spanish Agency of Medicines and Health Products (AEMPS). The AEMPS as a state agency attached to the Ministry of Health is a reference entity in Spain in charge of protecting public health through the strict regulation of medicines and health products. The AEMPS is considered highly reliable within the Spanish and European healthcare system as it operates under strict national and European regulations, and its procedures are aligned with the standards of the European Medicines Agency (EMA). Therefore, its data have a high degree of reliability.

− A paragraph explaining CIMA's reliability has been included in the methodology.

Discussion: According to FDA Pregnancy category and drugs -related risks information, generally chemotherapy and AA in specific are considered category X medication; where studies in animals or humans have demonstrated positive evidence of human fetal risk based on adverse reaction data from investigational or marketing experience, and the risks involved in use of the drug in pregnant women clearly outweigh potential benefits. Thus, no such information regarding the use of AA in pregnant and breast feeding, or its impact on infants exposure.

Indeed, there is no information on the use of antineoplastic agents in pregnant and breastfeeding women, nor on their impact on infant exposure, since being category X the risk clearly outweighs the possible benefits, they are not used = no data. To further clarify this information, we have expanded the paragraph in the discussion.

---

## [Decision Letter · Decision Letter 1]

7 Feb 2025

PONE-D-24-35201R1Population interest in and adequacy of the information on the safety of antineoplastic agents in the Spanish edition of WikipediaPLOS ONE

Dear Dr. Sanz-Valero,

Thank you for submitting your manuscript to PLOS ONE. After careful consideration, we feel that it has merit but does not fully meet PLOS ONE’s publication criteria as it currently stands. Therefore, we invite you to submit a revised version of the manuscript that addresses the points raised during the review process.

**ACADEMIC EDITOR: ** Dear Authors, the manuscript needs minor revisions, please respond point by point to the reviewers' requests.

Kind regards

We look forward to receiving your revised manuscript.

Kind regards,

Omar Enzo Santangelo

Academic Editor

PLOS ONE

Journal Requirements:

Reviewers' comments:

Reviewer's Responses to Questions

**Comments to the Author**

1. If the authors have adequately addressed your comments raised in a previous round of review and you feel that this manuscript is now acceptable for publication, you may indicate that here to bypass the “Comments to the Author” section, enter your conflict of interest statement in the “Confidential to Editor” section, and submit your "Accept" recommendation.

Reviewer #1: All comments have been addressed

Reviewer #2: All comments have been addressed

2. Is the manuscript technically sound, and do the data support the conclusions?

Reviewer #1: Yes

Reviewer #2: Yes

3. Has the statistical analysis been performed appropriately and rigorously? 

Reviewer #1: Yes

Reviewer #2: Yes

4. Have the authors made all data underlying the findings in their manuscript fully available?

Reviewer #1: Yes

Reviewer #2: Yes

5. Is the manuscript presented in an intelligible fashion and written in standard English?

Reviewer #1: Yes

Reviewer #2: Yes

6. Review Comments to the Author

Reviewer #1: ......................................................................................................

Reviewer #2: I think that the authors have fulfilled and well-explained all peer-reviewers remarks and notes on the basis of clinical and scientific tracks, and made all the recommended corrections punctually.

However, some efforts to change the abbreviation AA to ANP in the abstract section (at "Mansucript draft; page # 1) are required as being applied on the whole manuscript.

7. PLOS authors have the option to publish the peer review history of their article (what does this mean? ). If published, this will include your full peer review and any attached files.

**Do you want your identity to be public for this peer review?** For information about this choice, including consent withdrawal, please see our Privacy Policy .

Reviewer #1: No

Reviewer #2: No

---

## [Author Response · Author response to Decision Letter 2]

21 Feb 2025

Point-by-point response to the reviewer’s comments

First of all, we would like to thank you for the new review of our manuscript.

All changes made to this document are highlighted in bright colors in the text of the manuscript so you can easily see them.

Revisions carried out

All bibliographic references have been thoroughly reviewed, and no errors have been found nor have any retracted articles been cited.

Reviewer #2

I think that the authors have fulfilled and well-explained all peer-reviewers remarks and notes on the basis of clinical and scientific tracks, and made all the recommended corrections punctually.

However, some efforts to change the abbreviation AA to ANP in the abstract section (at "Mansucript draft; page # 1) are required as being applied on the whole manuscript.

In accordance with the reviewer's comments, all abbreviations 'AA' have been corrected and replaced with 'ANP'. Previously, they were highlighted in yellow, and in this latest review, they have been highlighted in green.

---

## [Editor Report · Decision Letter 2]

25 Feb 2025

Population interest in and adequacy of the information on the safety of antineoplastic agents in the Spanish edition of Wikipedia

PONE-D-24-35201R2

Dear Dr. Sanz-Valero,

We’re pleased to inform you that your manuscript has been judged scientifically suitable for publication and will be formally accepted for publication once it meets all outstanding technical requirements.

Kind regards,

Omar Enzo Santangelo

Academic Editor

PLOS ONE
---

## [Editor Report · Acceptance letter]

PONE-D-24-35201R2

PLOS ONE

Dear Dr. Sanz-Valero,

I'm pleased to inform you that your manuscript has been deemed suitable for publication in PLOS ONE. Congratulations! Your manuscript is now being handed over to our production team.

Kind regards,

on behalf of

Dr. Omar Enzo Santangelo

Academic Editor

PLOS ONE